# Weakly interacting Bose gas with two-body losses

Chang Liu[1], Zheyu Shi[2] and Ce Wang[3*]

**1** Institute for Advanced Study, Tsinghua University, Beijing 100084, China
**2** State Key Laboratory of Precision Spectroscopy,
East China Normal University, Shanghai 200062, China
**3** School of Physics Science and Engineering, Tongji University, Shanghai 200092, China

⋆ phywangce@gmail.com

## Abstract

We study the many-body dynamics of weakly interacting Bose gases with two-particle losses. We show that both the two-body interactions and losses in atomic gases may be tuned by controlling the inelastic scattering process between atoms by an optical Feshbach resonance. Interestingly, the low-energy behavior of the scattering amplitude is governed by a single parameter, i.e. the complex $s$-wave scattering length $a_c$. The many-body dynamics are thus described by a Lindblad master equation with complex scattering length. We solve this equation by applying the Bogoliubov approximation in analogy to the closed systems. Various peculiar dynamical properties are discovered, some of them may be regarded as the dissipative counterparts of the celebrated results in closed Bose gases. For example, we show that the next-order correction to the mean-field particle decay rate is to the order of $|na_c^3|^{1/2}$, which is an analogy of the Lee-Huang-Yang correction of Bose gases. It is also found that there exists a dynamical symmetry of symplectic group $Sp(4, \mathbb{C})$ in the quadratic Bogoliubov master equation, which is an analogy of the $SU(1,1)$ dynamical symmetry of the corresponding closed system. We further confirmed the validity of the Bogoliubov approximation by comparing its results with a full numerical calculation in a double-well toy model. Generalizations of other alternative approaches such as the dissipative version of the Gross-Pitaevskii equation and hydrodynamic theory are also discussed in the last.

 Check for updates

# 1   Introduction

Interacting Bose gas is a central topic in the fields of ultracold atoms and condensed matter physics, for it represents a paradigm of quantum many-body physics. Theoretically, if one focuses on the low-energy physics, the interactions between bosons can always be simplified by a zero-range one with a real $s$-wave scattering length $a_s$ that reproduces the same low-energy scattering phase shift [1], despite that the actual potentials are usually very complicated. Various many-body effects can then be studied theoretically with the help of the zero-range model [2, 3]. More importantly, experimental techniques such as Feshbach resonance can adjust $a$ along the real-axis in cold atomic gases [4], which allows us to demonstrate various many-body effects, from the Lee-Huang-Yang correction [5–9] for positive scattering length to the Bose nova effect [10–12] for negative scattering length.

Recently, with the development of theoretical and experimental methods [13–29], much more attention has been paid to behaviors of cold atom systems with dissipation. For bosonic and fermionic models, single particle and two-body dissipation processes such as particle pump and loss can be realized [30–36]. To better understand the many-body physics in open systems, it is then useful to introduce a zero-range model for Bose gases with two-body losses. In a previous work [37], the authors have shown that with a proper renormalization or regularization approach, the boson-boson interactions and losses can be effectively described by a complex contact interaction parameterized by a complex $s$-wave scattering length $a_c$. The physics of the dissipative Bose gases can then be regarded as an extension of $a_s$ from the real axis to the complex plane.

In this paper, we focus on the dissipative dynamics of weakly interacting Bose gas with two-body losses. Firstly, we show that complex scattering length $a_c$ can be effectively adjusted in the lower complex plane through optical Feshbach resonances. Then we study the many-body dynamics in the weakly interacting and dissipating region by introducing the Bogoliubov approximation [38]. Within the Bogoliubov approximation, the evolution process is governed by a quadratic time-dependent Lindblad equation. By numerically solving a toy model, we verify the accuracy of this approximation in open systems. Furthermore, we find within this approximation, the superoperators in the master equation form a closed algebra, which corresponds to a symplectic $\mathrm{Sp}(4, \mathbb{C})$ dynamical symmetry group, and helps to derive an exact solution for the evolution of density matrix. Our analysis also reveals that an $n$-mode bosonic system governed by a quadratic Lindbladian always has $\mathrm{Sp}(2n, \mathbb{C})$ dynamical symmetry. Finally, within the framework of the Keldysh path-integral method, we obtain a generalized non-Hermitian Gross-Pitaevskii equation which reproduces the same excitation spectrum as the Lindblad equation and leads to a set of dissipative hydrodynamic equations in the long wavelength limit.

The paper is organized as follows. In Section. 2, we show that the complex scattering length $a_c$ in an atomic gas can be tuned across the lower half complex plane via optical Feshbach resonances. Then in Section. 3 we introduce the single-channel master equation and the Bogoliubov approximation, which we use to solve the many-body properties of the dissipative

Bose gases. We verify the validity of the Bogoliubov approximation by numerically solving a toy model in Section. 4, and discuss the dynamical symmetry of the Bogoliubov Lindbladian in Section. 5. In Section. 6, we derive the dissipative version of the Gross-Pitaevskii equation and hydrodynamic theory using the Keldysh path integral formalism for open systems.

## 2 Tuning complex scattering lengths in experiments

In the scattering theory, it is known that the inelastic collisions between particles will cause two-body losses and eventually lead to a complex $s$-wave scattering length, which contains all the information of the low-energy scattering amplitude [1,39–41]. This suggests that one might control the two-particle interaction and losses experimentally by tuning the inelastic scattering process between atoms in a cold atomic gas. Experimentally, this tuning can be realized through the optical Feshbach resonance technique [42]. As shown in Fig. 1 (a), in the experiment, an external laser is applied to the atomic gas, and the frequency of the optical field is tuned close to a transition between two ground state atoms and a highly excited molecular state (i.e. a two-body bound state consists of a ground state atom and a highly excited atom). When the excited molecule spontaneously decays and emits a photon, the two atoms will be kicked out of the system because of the huge recoil momentum, thus causing two-particle losses in the atomic gas.

Theoretically, this process might be captured by a detailed calculation of the scattering amplitude for a finite-range multi-channel model, which shows that the complex scattering length is given by [39–41]

$$a_c(E) = a_{\text{bg}}\left(1 + \frac{\Gamma(I)}{E - \nu - \Gamma(I) + i\gamma/2}\right). \tag{1}$$

Here $\nu$ is the detuning of the laser field, $\gamma$ is the decay rate of the excited molecule, $a_{\text{bg}}$ is the (real) background $s$-wave scattering length, $E$ is the scattering energy, and $\Gamma(I)$ stands for the width of the resonance and is proportional to the intensity $I$ of the laser field.

In this chapter, we provide a simplified zero-range model to reproduce this formula and discuss the domains on the complex plane where $a_c$ might reach using optical Feshbach resonances. To construct the model, we first introduce the bosonic field operator $\hat{b}$ ($\hat{d}$) for the ground state atoms (excited molecules), and the Hamiltonian of the system may be written as ($\hbar$ is set to unity in this paper)

$$\hat{H}_{\text{ofr}} = \sum_{\mathbf{k}}\left(\epsilon_{\mathbf{k}}\hat{b}_{\mathbf{k}}^{\dagger}\hat{b}_{\mathbf{k}} + \xi_{\mathbf{k}}\hat{d}_{\mathbf{k}}^{\dagger}\hat{d}_{\mathbf{k}}\right) + \frac{g}{2\Omega}\sum_{\mathbf{k},\mathbf{k}',\mathbf{p}}\hat{b}_{\mathbf{k}+\mathbf{p}}^{\dagger}\hat{b}_{\mathbf{k}'-\mathbf{p}}^{\dagger}\hat{b}_{\mathbf{k}'}\hat{b}_{\mathbf{k}}$$
$$+ \frac{1}{\sqrt{\Omega}}\sum_{\mathbf{k},\mathbf{p}}\left(\alpha\hat{d}_{\mathbf{p}}^{\dagger}\hat{b}_{\frac{\mathbf{p}}{2}-\mathbf{k}}\hat{b}_{\frac{\mathbf{p}}{2}+\mathbf{k}} + \alpha^*\hat{d}_{\mathbf{p}}\hat{b}_{\frac{\mathbf{p}}{2}-\mathbf{k}}^{\dagger}\hat{b}_{\frac{\mathbf{p}}{2}+\mathbf{k}}^{\dagger}\right),$$

where $g$ is the interacting strength between atoms in the open channel, $\epsilon_{\mathbf{k}} = \frac{k^2}{2m}$ and $\xi_{\mathbf{k}} = \frac{k^2}{4m} + \nu$, $m$ is the atom mass, $\Omega$ is the volume. The two channels are coupled by laser light with strength $\alpha$.

The spontaneous radiation of the bound-state molecule can be characterized by the following Lindblad master equation [43]

$$\partial_t\hat{\rho} = \mathcal{L}(\hat{\rho}) = -i[\hat{H}_{\text{ofr}}, \hat{\rho}] - \frac{\gamma}{2}\sum_{\mathbf{k}}\{\hat{d}_{\mathbf{k}}^{\dagger}\hat{d}_{\mathbf{k}}, \hat{\rho}\} + \gamma\sum_{\mathbf{k}}\hat{d}_{\mathbf{k}}\hat{\rho}\hat{d}_{\mathbf{k}}^{\dagger}.$$

For a two-body process, the dynamics of the corresponding density matrix (in two-particle

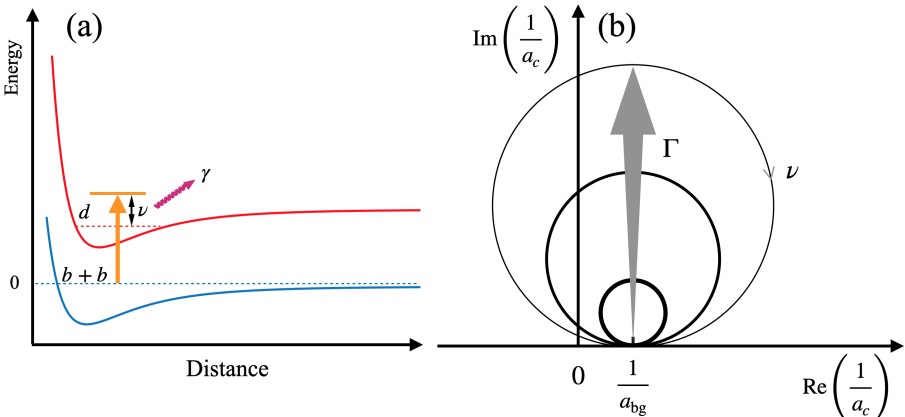

Figure 1: (a)The schematic for the optical Feshbach resonance model. In the open channel, a pair of atoms $b$ is interacting via strength $g$. They are coupled to a molecule state $d$ in the closed channel by laser light at detuning $\nu$. The molecular state spontaneously radiates at rate $\gamma$. (b) The range of $a_c^{-1}(0)$. For fixed $\Gamma(I)$, the trajectory of $a_c^{-1}(0)$ forms a circle tangent to the real-axis at $a_{\rm bg}^{-1}$ with radius $\frac{\Gamma(I)}{\gamma a_{\rm bg}}$ in the upper half complex plane when changing $\nu$ within $(-\infty, +\infty)$. Then the radius of this circle will increase as we gradually turn on $\Gamma(I)$.

Fock space) is equivalent to the evolution under a non-hermitian Hamiltonian

$$\hat{H}_{\text{2-body}} = \hat{H}_{\text{ofr}} - i\frac{\gamma}{2}\sum_{\mathbf{k}} \hat{d}_{\mathbf{k}}^{\dagger}\hat{d}_{\mathbf{k}}. \tag{2}$$

We can obtain the two-body scattering matrix $T_2$ for this $\hat{H}_{\text{eff}}$ at relative kinetic energy $E$

$$T_2(E) = \left(\left(g + \frac{|\alpha|^2}{E - \nu + i\gamma/2}\right)^{-1} - \frac{1}{\Omega}\sum_k \frac{1}{E - k^2/m}\right)^{-1}.$$

By matching the derived T-matrix with the usual low-energy expansion formula $\frac{4\pi}{m}\left(\frac{1}{a_c(E)} + i\sqrt{mE}\right)^{-1}$ [1], we obtain the renormalize relation for the complex scattering length $a_c$,

$$\frac{m}{4\pi a_c(E)} = \left(g + \frac{|\alpha|^2}{E - \nu + i\gamma/2}\right)^{-1} + \frac{1}{\Omega}\sum_{|\mathbf{k}|<\Lambda} \frac{m}{k^2}, \tag{3}$$

where $\Lambda$ is the momentum cut-off and $a_c(E)$ can be further written as

$$a_c(E) = a_{\rm bg} + \frac{m}{4\pi}\frac{|\alpha_{\rm re}|^2}{E - \nu_{\rm re} + i\gamma/2}, \tag{4}$$

with $\frac{m}{4\pi a_{\rm bg}} = \frac{1}{g} + \frac{m\Lambda}{2\pi^2}$, $|\alpha_{\rm re}|^2 = \frac{|\alpha|^2}{(1 + mg\Lambda/(2\pi^2))^2}$, $\nu_{\rm re} = \nu - \frac{m\Lambda|\alpha|^2}{2\pi^2 + mg\Lambda}$.

The above formula is exactly eq. (1) if we take the limit $\Lambda \to \infty$, and write $\Gamma(I) = \frac{m|\alpha_{\rm re}|^2}{4\pi a_{\rm bg}}$.[1]

For a dilute gas at extremely low temperature, the kinetic energy $E$ is negligible and the many-body effect can be well captured by the zero energy scattering length $a_c(E=0)$. A general $a_c(0)$ can be achieved via changing the detuning $\nu$ and density $I$. To be more clear, as

---

[1]This term could be regarded as $\Gamma(I)$ because the square of the renormalized coupling strength $|\alpha_{\rm re}|^2$ is also proportional to the laser intensity.

shown in Fig. 1(b), at fixed $\Gamma(I)$, the trajectory of $a_c^{-1}(E=0)$ form a circle tangent to the real-axis at $a_{\mathrm{bg}}^{-1}$ with radius $\frac{\Gamma(I)}{\gamma a_{\mathrm{bg}}}$ in the upper half complex when tuning $\nu \in (-\infty, +\infty)$. Therefore, as we gradually increase $\Gamma(I)$, this circle can sweep across the entire upper half-plane.

Here we comment on the difference between tuning complex scattering length and real scattering length. The real scattering length can also be controlled in a magnetic-induced Feshbach resonance by varying the strength of the magnetic field. However, to achieve a general complex scattering length, we need at least two adjustable parameters. Thus we choose the optical resonance model rather than the magnetic Feshbach resonance.

## 3 Dissipative Bose gas

For non-dissipative atomic gases, it is known that the interaction between particles may be described by a single-channel contact potential, provided that the Feshbach resonance is "wide", i.e. that the occupation in the closed molecular channel is small compared to the open scattering channel [41]. The Hermitian Hamiltonian of a Bose gas may be written as

$$\hat{H} = \sum_{\mathbf{k}} \epsilon_{\mathbf{k}} \hat{a}_{\mathbf{k}}^\dagger \hat{a}_{\mathbf{k}} + \frac{g}{2\Omega} \sum_{\mathbf{k},\mathbf{k}',\mathbf{p}} \hat{a}_{\mathbf{k}+\mathbf{p}}^\dagger \hat{a}_{\mathbf{k}'-\mathbf{p}}^\dagger \hat{a}_{\mathbf{k}'} \hat{a}_{\mathbf{k}}, \tag{5}$$

where $\hat{a}_{\mathbf{k}}$ is the annihilation operator for the atoms with momentum $\mathbf{k}$, $\epsilon_{\mathbf{k}}$ is the kinetic energy, $g$ is the coupling constant, $\Omega$ is the system volume.

The single-channel Hamiltonian greatly simplifies the calculation of the many-body properties in closed Bose gases. Thus it is desirable to construct a single-channel model to describe the many-body dynamics of dissipative atomic gases with complex scattering lengths. In the previous work [37], the authors have studied this problem and constructed a renormalizable single-channel model for systems across a "wide" optical Feshbach resonance. In this work, we focus on the many-body dynamics of this model, and the study of systems across "narrow" optical Feshbach resonances will be pursued in the future. In the single-channel model, the open system dynamics are governed by the master equation [37,43]

$$\partial_t \hat{\rho} = -i(\hat{H}_{\mathrm{eff}} \hat{\rho} - \hat{\rho} \hat{H}_{\mathrm{eff}}^\dagger) + \mathcal{J} \hat{\rho}, \tag{6}$$

with the non-Hermitian effective Hamiltonian

$$\hat{H}_{\mathrm{eff}} = H - \frac{i\gamma_b}{2\Omega} \sum_{\mathbf{k},\mathbf{k}',\mathbf{p}} \hat{a}_{\mathbf{k}+\mathbf{p}}^\dagger \hat{a}_{\mathbf{k}'-\mathbf{p}}^\dagger \hat{a}_{\mathbf{k}'} \hat{a}_{\mathbf{k}}, \tag{7}$$

and the recycling term

$$\mathcal{J} \hat{\rho} = \frac{\gamma_b}{\Omega} \sum_{\mathbf{k},\mathbf{k}',\mathbf{p}} \hat{a}_{\mathbf{k}'} \hat{a}_{\mathbf{k}} \hat{\rho} \hat{a}_{\mathbf{k}'-\mathbf{p}}^\dagger \hat{a}_{\mathbf{k}+\mathbf{p}}^\dagger. \tag{8}$$

This master equation describes the dynamics of Bose atoms, denoted by $\hat{a}$, as they interact with each other through a bare coupling constant $g_b$ while decaying to the environment (i.e. no longer confined by the trapping potential) via a two-body losses process characterized by a bare coupling constant $\gamma_b$. To regularize the contact interaction, we can use the renormalization relation [37]

$$\frac{1}{g_b - i\gamma_b} = \frac{1}{g - i\gamma} - \frac{1}{\Omega} \sum_{\mathbf{k}} \frac{1}{2\epsilon_{\mathbf{k}}}, \tag{9}$$

and define $g - i\gamma \equiv \frac{4\pi a_c}{m}$ as renormalized complex coupling constant.

## Bogoliubov approximation

To solve the spectrum of isolated weakly interacting Bose gas, we can use the Bogoliubov approximation and diagonalize the quadratic Bogoliubov Hamiltonian. Similarly, we can generalize Bogoliubov's transformation in the open system. Starting from a condensate initial state and assuming the condensate part is always much larger than quantum depletion during the time evolution, we can trace out the condensed part by replacing all the $\hat{a}_{\mathbf{0}}, \hat{a}_{\mathbf{0}}^{\dagger}$ with the square root of the zero-momentum atom number $N - \sum_{\mathbf{k} \neq \mathbf{0}} \hat{a}_{\mathbf{k}}^{\dagger} \hat{a}_{\mathbf{k}}$, then we obtain an approximate master equation for the reduced density matrix $\hat{\rho}'$ as $\partial_t \hat{\rho}' \approx \mathcal{L}_B(\hat{\rho}')$ with

$$\mathcal{L}_B(\hat{\rho}') = -i[\hat{H}_B, \hat{\rho}'] - 2\gamma n \sum_{\mathbf{k} \neq \mathbf{0}} \{\hat{a}_{\mathbf{k}}^{\dagger} \hat{a}_{\mathbf{k}}, \hat{\rho}'\} + 4\gamma n \sum_{\mathbf{k} \neq \mathbf{0}} \hat{a}_{\mathbf{k}} \hat{\rho}' \hat{a}_{\mathbf{k}}^{\dagger}, \tag{10}$$

and

$$\hat{H}_B = \frac{gnN}{2} + \sum_{\mathbf{k} \neq \mathbf{0}} \left( (\epsilon_k + gn)\hat{a}_{\mathbf{k}}^{\dagger} \hat{a}_{\mathbf{k}} + \frac{gn - i\gamma n}{2} \hat{a}_{\mathbf{k}}^{\dagger} \hat{a}_{-\mathbf{k}}^{\dagger} + h.c. \right),$$

where $N$ is the total atom number and $n = \frac{N}{\Omega}$ is the density. Here we use renormalized parameters $g, \gamma$ to replace the bare parameters $g_b, \gamma_b$ like in conventional Bogoliubov Hamiltonian [38]. Different from the time-independent Bogoliubov Hamiltonian in a closed system, Lindbladian after Bogoliubov transformation is time-dependent due to the two-body losses. The density $n$ is decreasing as a function of time $t$, at mean-field level, we have $n(t) = n_0(1 + 2\gamma n_0 t)^{-1}$.

The conventional Bogoliubov Hamiltonian is the linear combination of three operators $\hat{A}_{\mathbf{k}}^0 = \frac{1}{2}(\hat{N}_{\mathbf{k}} + \hat{N}_{-\mathbf{k}} + 1)$, $\hat{A}_{\mathbf{k}}^1 = \frac{1}{2}(\hat{a}_{\mathbf{k}}^{\dagger} \hat{a}_{-\mathbf{k}}^{\dagger} + h.c.)$ and $\hat{A}_{\mathbf{k}}^2 = \frac{1}{2i}(\hat{a}_{\mathbf{k}}^{\dagger} \hat{a}_{-\mathbf{k}}^{\dagger} - h.c.)$, these operators form a closed SU(1,1) algebra [44–46],

$$\left[\hat{A}_{\mathbf{k}}^1, \hat{A}_{\mathbf{k}}^2\right] = -i\hat{A}_{\mathbf{k}}^0, \left[\hat{A}_{\mathbf{k}}^2, \hat{A}_{\mathbf{k}}^0\right] = i\hat{A}_{\mathbf{k}}^1, \left[\hat{A}_{\mathbf{k}}^0, \hat{A}_{\mathbf{k}}^1\right] = i\hat{A}_{\mathbf{k}}^2. \tag{11}$$

As a result, for an isolated system, the Hamiltonian has SU(1,1) dynamical symmetry, and the Heisenberg equations of these three operators are closed. When introducing the two-body losses, the Heisenberg equations for these three operators are no longer closed. However, the evolution of their expectation values $\mathbf{A} = (\langle A_0^{\mathbf{k}} \rangle, \langle A_2^{\mathbf{k}} \rangle, \langle A_2^{\mathbf{k}} \rangle)^{\mathrm{T}}$ are still closed, satisfying

$$\dot{\mathbf{A}} = -2 \begin{pmatrix} 2\gamma n & \gamma n & -gn \\ \gamma n & 2\gamma n & \epsilon_{\mathbf{k}} + gn \\ -gn & -\epsilon_{\mathbf{k}} - gn & 2\gamma n \end{pmatrix} \mathbf{A} + \begin{pmatrix} 2\gamma n \\ 0 \\ 0 \end{pmatrix}. \tag{12}$$

Here, we focused on the short-time dynamics where $\gamma nt \ll 1$ such that we may approximate $n$ as a constant at a fixed time t. We can diagonalize the $3 \times 3$ matrix in eq. (12) to obtain the eigenvalue $-2i\xi_{\mathbf{k},i}$, where the quasi-steady state eigenvalue is

$$\xi_{\mathbf{k},0} = -2i\gamma n, \quad \xi_{\mathbf{k},(1,2)} = -2i\gamma n \pm \sqrt{\epsilon_{\mathbf{k}}^2 + 2g_0 n \epsilon_{\mathbf{k}} - \gamma_0^2 n^2}.$$

A phase boundary between stable and unstable BEC can be obtained by this quasi-steady state eigenvalue [37].

## Lee-Huang-Yang correction

Furthermore, we will give a detailed derivation for the Lee-Huang-Yang correction [47,48] for this open system. Consider quasi-steady-state eigenvalue, up to $O(\gamma nt)$, simple solution of eq. (12) can be obtained

$$\mathbf{A}(t) = \mathbf{A}_s + e^{-4\gamma nt} (\mathbf{B} + \mathbf{C} \cos 2\xi_{\mathbf{k}} t + \mathbf{D} \sin 2\xi_{\mathbf{k}} t), \tag{13}$$

constant vectors $\mathbf{B}, \mathbf{C}, \mathbf{D}$ can be determined by the initial value of $\mathbf{A}$, and

$$\mathbf{A}_s^{\mathbf{T}} = \begin{pmatrix} \frac{1}{2} + \frac{(g^2+\gamma^2)n^2}{2\epsilon_{\mathbf{k}}^2 + 4gn\epsilon_{\mathbf{k}} + 6\gamma^2 n^2} \\ -\frac{gn\epsilon_{\mathbf{k}} + g^2 n^2 + 2\gamma^2 n^2}{2\epsilon_{\mathbf{k}}^2 + 4gn\epsilon_{\mathbf{k}} + 6\gamma^2 n^2} \\ -\frac{\gamma n(\epsilon_{\mathbf{k}} - gn)}{2\epsilon_{\mathbf{k}}^2 + 4gn\epsilon_{\mathbf{k}} + 6\gamma^2 n^2} \end{pmatrix}. \tag{14}$$

We see that $\mathbf{A}$ never reaches the steady value $\mathbf{A}_s$ because the asymptotic solution eq. (13) only works for time interval $0 \le t \ll \hbar/\gamma n$. Nevertheless, it is still useful to calculate some physical properties for this steady state, since this helps to verify the renormalization relation given in eq. (9).

When $\mathbf{A}$ reaches this steady value, we see that $n_{\mathbf{k}} + n_{-\mathbf{k}} = \langle A_0^{\mathbf{k}} \rangle - \frac{1}{2}$ also becomes steady. Thus we get the total particle losses rate $\frac{dN}{dt} = \frac{dN_0}{dt}$. And we may calculate $dN_0/dt$ by

$$\frac{dN_0}{dt} = \partial_t \text{tr}\left(\hat{\rho} a_0^\dagger a_0\right) = \text{tr}\left(\mathcal{L}(\hat{\rho}) a_0^\dagger a_0\right) = \text{tr}\left(\hat{\rho}\mathcal{L}'(a_0^\dagger a_0)\right),$$

with $\mathcal{L}'$ defined by

$$\mathcal{L}'(\hat{O}) \equiv i\left[\hat{H}, \hat{O}\right] - \frac{\gamma}{2V}\sum_{\mathbf{k},\mathbf{k}',\mathbf{p}} \left\{a_{\mathbf{k}+\mathbf{p}}^\dagger a_{\mathbf{k}'-\mathbf{p}}^\dagger a_{\mathbf{k}'} a_{\mathbf{k}}, \hat{O}\right\} + \frac{\gamma}{2V}\sum_{\mathbf{k},\mathbf{k}',\mathbf{p}} a_{\mathbf{k}+\mathbf{p}}^\dagger a_{\mathbf{k}'-\mathbf{p}}^\dagger \hat{O} a_{\mathbf{k}'} a_{\mathbf{k}}.$$

It is then straightforward to show that

$$\mathcal{L}'(a_0^\dagger a_0) \simeq -2\gamma nN - 2\sum_{\mathbf{k}\neq 0}\left(\gamma n A_1^{\mathbf{k}} + gn A_2^{\mathbf{k}}\right). \tag{15}$$

This leads to

$$\frac{dN_0}{dt} = -2\gamma nN - 2\sum_{\mathbf{k}\neq 0}\left(\gamma n \langle A_1^{\mathbf{k}} \rangle + gn \langle A_2^{\mathbf{k}} \rangle\right). \tag{16}$$

From this equation, we see that the first term corresponds to the mean-field decay which leads to $n \simeq n(1 + 2\gamma nt)^{-1}$ as mentioned previously. After inserting the steady value, we have

$$\frac{dN_0}{dt} = -2\gamma nN + 2\gamma n\sum_{\mathbf{k}} \frac{gn\epsilon_{\mathbf{k}} + \gamma^2 n^2}{\epsilon_{\mathbf{k}}^2 + 2gn\epsilon_{\mathbf{k}} + 3\gamma^2 n^2}. \tag{17}$$

We immediately find that the summation diverges for large momentum. Similar divergence occurs in the calculation for ground state energy of Bose gas in a closed system [49, 50]. This divergence arises from the fact that the renormalized interaction is only valid for small momenta. To cure this divergence, one can introduce an intermediate momentum cutoff and then consider an effective interaction with second-order processes in the mean-field energy component. For our case, the second-order effective interaction parameter $\tilde{g}$ and $\tilde{\gamma}$ is given by

$$\tilde{g} - i\tilde{\gamma} = (g - i\gamma)\left(1 + \frac{g - i\gamma}{V}\sum_{\mathbf{k}}\frac{1}{2\epsilon_{\mathbf{k}}}\right). \tag{18}$$

We thus obtain

$$\tilde{\gamma} = \gamma + \frac{g\gamma}{V}\sum_{\mathbf{k}}\frac{1}{\epsilon_{\mathbf{k}}}. \tag{19}$$

Replacing $\gamma$ in the mean-field part of eq. (17) by $\tilde{\gamma}$, we have

$$
\begin{aligned}
\frac{dN}{dt} &= \frac{dN_0}{dt} \\
&= -2\gamma n N + 2\gamma n \sum_{\mathbf{k}} \left( \frac{gn\epsilon_{\mathbf{k}} + \gamma^2 n^2}{\epsilon_{\mathbf{k}}^2 + 2gn\epsilon_{\mathbf{k}} + 3\gamma^2 n^2} - \frac{gn}{\epsilon_{\mathbf{k}}} \right) \\
&= -2\gamma n N \left[ 1 + c_\theta (n|a_c|^3)^{1/2} \right],
\end{aligned}
\tag{20}
$$

with $c_\theta$ a constant that only depends on the argument of the complex scattering length $a_c$,

$$
c_\theta = 4\sqrt{2\pi} \left( \frac{\cos 2\theta}{\sqrt{\cos(\theta - \pi/3)}} + 2\cos\theta \sqrt{\cos(\theta - \pi/3)} \right).
$$

Here, $\theta$ is defined as $\theta = -\arg(a_c) = -\arg(a_r + ia_i) \in [0, \pi]$, where $a_c$ is expressed as two real parameters: $a_c = a_r + ia_i$.

We note that the $n|a_c|^3$ term in eq. (20) is an analog of the celebrated Lee-Huang-Yang correction to the ground state energy of a weakly interacting Bose gas [47, 48],

$$
E_0 = \frac{g_0 n N}{2} \left[ 1 + \frac{128}{15\pi^{1/2}} (na_r^3)^{1/2} + O(n \log n) \right].
\tag{21}
$$

With the help of renormalization relation, we can eliminate the divergence eq. (16) and obtain a physical result of particle loss rate.

It is well known that, for Bose gas without a two-particle loss ($a_i = 0$), the system is dynamically unstable when $a_r < 0$. This instability is reflected in eq. (21), which becomes ill-defined for a negative real scattering length. Similarly, eq. (20) reflects certain dynamic instability in open systems. One can check that the coefficient $c_\theta$ becomes ill-defined for $\theta \in [5\pi/6, \pi]$, suggesting that our approach is not valid in this regime. This is because the quantum depletion (Bogoliubov modes) grows rapidly in this regime, which invalidates the assumption that $\hat{a}_{\mathbf{0}} \approx \hat{a}_{\mathbf{0}}^\dagger \approx \sqrt{N_0}$ [37].

## 4 A toy model

Our calculation is based on the assumption that the Bogoliubov approximation is always valid during the dynamical evolution. It is impossible to verify this approximation numerically in the thermodynamic limit due to the exponential growth of the Hilbert space dimension in the quantum many-body system. To numerically compare the dynamics process generated by the original Lindbladian and the Bogoliubov Lindbladian, we introduce a toy model that can capture the interacting and dissipation features of the open bosonic system.

**Numerical model**

As shown in Fig. 2, we consider a double-well model, the annihilation operator for the bosonic mode in the right (left) well is denoted by $\hat{b}_r(\hat{b}_l)$. The evolution of this system is governed by the master equation

$$
\partial_t \hat{\rho} = -i[\hat{H}_{\mathrm{dw}}, \hat{\rho}] - \gamma \sum_{i=r,l} \left( \{\hat{\Gamma}_i^\dagger \hat{\Gamma}_i, \rho\} + 2\hat{\Gamma}_i \rho \hat{\Gamma}_i^\dagger \right),
\tag{22}
$$

where the Hamiltonian is

$$
\hat{H}_{\mathrm{dw}} = -t \left( \hat{b}_r^\dagger \hat{b}_l + \hat{b}_l^\dagger \hat{b}_r \right) + g \left( \hat{b}_r^\dagger \hat{b}_r^\dagger \hat{b}_r \hat{b}_r + \hat{b}_l^\dagger \hat{b}_l^\dagger \hat{b}_l \hat{b}_l \right),
\tag{23}
$$

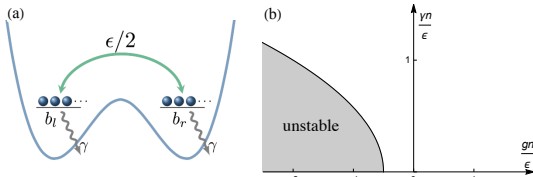

Figure 2: (a) Schematic of a double-well toy model with interacting and two-body losses, $\epsilon$ represent the energy detuning. (b) The phase diagram of double-well model on $\frac{\gamma n}{\epsilon} - \frac{gn}{\epsilon}$ plane. When $\frac{gn}{\epsilon} < -\frac{1+3(\frac{\gamma n}{\epsilon})^2}{2}$, quasi-steady state eigenvalue has a positive imaginary part, which represents the exponential growth of non-condensate particle number, the system is in the unstable phase. When $\frac{gn}{\epsilon} > -\frac{1+3(\frac{\gamma n}{\epsilon})^2}{2}$, the number of the minority particle is always much smaller than the condensate number during the time evolution, the system is in the stable phase.

with $t$ the hopping strength between the two wells, $g$ the on-site interaction strength. The Lindblad operators for the onsite two-body losses are $\hat{\Gamma}_r = \hat{b}_r \hat{b}_r, \hat{\Gamma}_l = \hat{b}_l \hat{b}_l$, with $\gamma$ the decay rate.

Corresponding to the Bose gas model, we consider the "zero momentum" mode $\hat{a}_0 = \frac{1}{\sqrt{2}}(\hat{b}_r + \hat{b}_l)$ as majority part while the "non-zero momentum" mode $\hat{a}_1 = \frac{1}{\sqrt{2}}(\hat{b}_r - \hat{b}_l)$ as small depletion part. We can shift the energy for the mode $\hat{a}_0$ to zero without loss of generality and the mode $\hat{a}_1$ has a detuning $\epsilon = 2t$ from the mode $\hat{a}_0$. Omitting the interacting or losses terms only including minority part, such as $\hat{a}_1^\dagger \hat{a}_1^\dagger \hat{a}_1 \hat{a}_1$, the master equation is then given by

$$\partial_t \hat{\rho} = -i\left(\hat{H}_a^{\mathrm{eff}}\hat{\rho} - \hat{\rho}\hat{H}_a^{\mathrm{eff}\dagger}\right) + \mathcal{J}\hat{\rho}, \tag{24}$$

where the effective Hamiltonian $\hat{H}_a^{\mathrm{eff}}$ for $\hat{a}_0, \hat{a}_1$ is

$$\hat{H}_a^{\mathrm{eff}} = \epsilon \hat{a}_1^\dagger \hat{a}_1 + \frac{g-i\gamma}{2}\left(\hat{a}_0^\dagger \hat{a}_0^\dagger \hat{a}_0 \hat{a}_0 + \hat{a}_0^\dagger \hat{a}_0^\dagger \hat{a}_1 \hat{a}_1 + \hat{a}_1^\dagger \hat{a}_1^\dagger \hat{a}_0 \hat{a}_0 + 4\hat{a}_0^\dagger \hat{a}_0 \hat{a}_1^\dagger \hat{a}_1\right), \tag{25}$$

and the recycling term in eq. (24) are similar to Lindbladian of Bose gas with two-body losses,

$$\mathcal{J}\hat{\rho} = \gamma\left(\hat{a}_0 \hat{a}_0 \hat{\rho} \hat{a}_0^\dagger \hat{a}_0^\dagger + \hat{a}_1 \hat{a}_1 \hat{\rho} \hat{a}_0^\dagger \hat{a}_0^\dagger + \hat{a}_0 \hat{a}_0 \hat{\rho} \hat{a}_1^\dagger \hat{a}_1^\dagger + 4\hat{a}_0 \hat{a}_1 \hat{\rho} \hat{a}_0^\dagger \hat{a}_1^\dagger\right).$$

In this model, the effective Hamiltonian $\hat{H}_a^{\mathrm{eff}}$ conserves the particle number $\hat{n} = \hat{a}_0^\dagger \hat{a}_0 + \hat{a}_1^\dagger \hat{a}_1$ while the recycling term always annihilates two particles, thus the master equation can be decomposed to a series of hierarchy equations for $\hat{\rho}_{i,j}$ [37]

$$\partial_t \hat{\rho}_{i,j} = -i\left(\hat{H}_a^{\mathrm{eff}}\hat{\rho}_{i,j} - \hat{\rho}_{i,j}\hat{H}_a^{\mathrm{eff}\dagger}\right) + \mathcal{J}\hat{\rho}_{i+2,j+2}, \tag{26}$$

where $\hat{\rho}_{i,j} = \hat{P}_i \hat{\rho} \hat{P}_j$, with $\hat{P}_i$ the projection operator for the $i$-particle Fock space. Then for the dynamical evolution of eq. (24) with initial density matrix a pure state of particle number $N$, only the projected density matrix $\hat{\rho}_{N,N}$, $\hat{\rho}_{N-2,N-2}, \hat{\rho}_{N-4,N-4}$... is involved and the numerical simulation is performed by calculating these differential equations.

Now considering the Bogoliubov approximation, we replace the $\hat{a}_0$ and $\hat{a}_0^\dagger$ with the square root of the condensate atoms number $n - \hat{a}_1^\dagger \hat{a}_1$, then we can obtain the approximated Lindbladian as

$$\begin{aligned}
\mathcal{L}_{\mathrm{dw}}^B \hat{\rho} = &-i\left[\hat{H}_{\mathrm{dw}}^B, \hat{\rho}\right] - \frac{\gamma}{2}\left\{n\hat{a}_1 \hat{a}_1 + n\hat{a}_1^\dagger \hat{a}_1^\dagger + 4n\hat{a}_1^\dagger \hat{a}_1, \hat{\rho}\right\} \\
&+ \gamma\left(n\hat{a}_1 \hat{a}_1 \hat{\rho} + n\hat{\rho}\hat{a}_1^\dagger \hat{a}_1^\dagger + 4n\hat{a}_1 \hat{\rho} \hat{a}_1^\dagger\right),
\end{aligned} \tag{27}$$

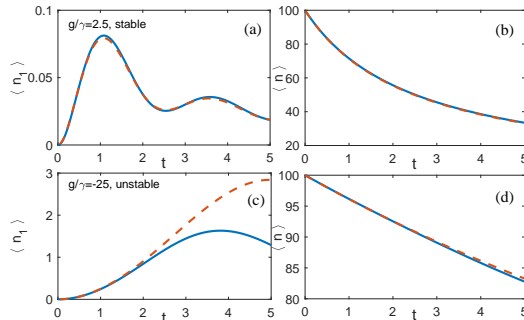

Figure 3: Quench dynamics under time evolution using two methods with total particle number $N = 100$ and energy in unit of $\epsilon$. (solid: exact time evolution governed by Lindbladian eq. (24); dashed: approximated Lindbladian eq. (27)). The initial condition is all particles condensate at groundstate, $n_0 = N$. (a),(b)System at stable phase with coupling strength $gN = 0.52\epsilon$ and dissipation strength $\gamma N = 0.20\epsilon$. (c),(d) System at stable phase with $gN = -0.52\epsilon$ and $\gamma N = 0.02\epsilon$.

where the Hamiltonian $\hat{H}_{dw}^B$ is

$$\hat{H}_{dw}^B = \epsilon \hat{a}_1^\dagger \hat{a}_1 + \frac{g}{2}(n^2 + n\hat{a}_1\hat{a}_1 + n\hat{a}_1^\dagger\hat{a}_1^\dagger + 2n\hat{a}_1^\dagger\hat{a}_1), \tag{28}$$

and time-dependent particle number is given by $n = N(1 + 2\gamma Nt)^{-1}$. Similar to eq. (12) in the Bose gas case, the expectation value of the three SU(1,1) operators $\hat{A}_{dw}^1 = (\hat{a}_1^\dagger\hat{a}_1 + \hat{a}_1\hat{a}_1^\dagger)/2, \hat{A}_{dw}^2 = (\hat{a}_1^\dagger\hat{a}_1^\dagger + \hat{a}_1\hat{a}_1)/2$ and $\hat{A}_{dw}^3 = (\hat{a}_1^\dagger\hat{a}_1^\dagger - \hat{a}_1\hat{a}_1)/2i$ form a closed set of differential equations,

$$\dot{\mathbf{A}}_{dw} = -2\begin{pmatrix} 2\gamma n & \gamma n & -gn \\ \gamma n & 2\gamma n & \epsilon + gn \\ -gn & -\epsilon - gn & 2\gamma n \end{pmatrix}\mathbf{A}_{dw} + \begin{pmatrix} \gamma n \\ 0 \\ 0 \end{pmatrix}, \tag{29}$$

with $\mathbf{A}_{dw} = (\langle\hat{A}_{dw}^1\rangle, \langle\hat{A}_{dw}^2\rangle, \langle\hat{A}_{dw}^3\rangle)^T$. The three eigenvalues $-2i\xi_i$ for the $3 \times 3$ matrix in eq. (29) are

$$\xi_0 = -2i\gamma n, \quad \xi_{(1,2)} = -2i\gamma n \pm \sqrt{\epsilon^2 + 2gn\epsilon - \gamma^2 n^2}. \tag{30}$$

When the imaginary part of $\xi_1$ is larger than zero, the corresponding eigenvalue becomes a positive number, thus the particle number $\hat{n}_1$ grows exponentially in short time leading to an unstable dynamic. The phase boundary[2] for this stability is given by $\frac{gn}{\epsilon} = -\frac{1+3(\frac{\gamma n}{\epsilon})^2}{2}$ as shown in Fig. 2(b).

## Numerical results

We now check the accuracy of the Bogoliubov approximation by comparing the numerical simulation result from eq. (24) and eq. (27). The evolution begins with an initial state with $N = 100$ particles occupying the mode $\hat{a}_0$. As shown in Fig. 3, the two evolution results almost coincide in the stable phase. While in the unstable phase, due to the increase of the quantum depletion, the particle number $\hat{n}_1$ predicted by the Bogoliubov approximation eq. (27) deviates from the exact time evolution governed by eq. (24) at long time. This result confirms that our Bogoliubov approximation for the Lindblad master equation is valid as long as the ratio between the density of quantum depletion to the total density is much smaller than the unitary.

---

[2]In fact, the transition between stable and unstable dynamics is a crossover since the position of the "boundary" depends on the density which decreases during the evolution.

# 5 Dynamical symmetry of Lindbladian

In this section, we discuss the dynamical symmetry of quadratic Lindbladians in detail, which can greatly simplify the calculation of the master eq. (10). It is worth mentioning that a time-independent quadratic Lindbladian may be solved explicitly using the third quantization method [51,52]. However, this cannot be directly applied to our problem because of the time-dependent $n(t)$ contained in the master eq. (10). Fortunately, in our Bogoliubov Lindbladian, it can be shown that, besides the obvious $U(1)$ and $Z_2$ symmetry, the algebraic structures of the superoperators defined on the density matrix space also contain a hidden symplectic $\text{Sp}(4,\mathbb{C})$ dynamical symmetry. This dynamical symmetry provides a relatively simple way [53, 54] to construct exact solutions to the time-dependent Lindbladian algebraically.

**Symmetry**

The Lindbladian $\mathcal{L}_B$ in eq. (10) has two obvious symmetries, i.e. an extended $U(1)$ symmetry and a $Z_2$ symmetry. To see the $U(1)$ symmetry, one only needs to verify that the Lindbladian is invariant under transformation $\hat{a}_{\mathbf{k}} \to \hat{a}_{\mathbf{k}} e^{i\phi}, \hat{a}_{\mathbf{k}}^\dagger \to \hat{a}_{\mathbf{k}}^\dagger e^{-i\phi}$. The corresponding conserved superoperator is thus $\tilde{Q} = [\hat{N}, \circ]$. While for the $Z_2$ symmetry, it can be verified by realizing $\mathcal{L}_B$ is invariant under $\hat{a}_{\mathbf{k}} \to -\hat{a}_{\mathbf{k}}$ or $\hat{a}_{\mathbf{k}}^\dagger \to -\hat{a}_{\mathbf{k}}^\dagger$.

However, these two symmetries alone are not enough to obtain an analytical solution to the master eq. (10). To solve the quench dynamics governed by this time-dependent master equation, we need to generalize the concept of dynamical symmetry to open systems, i.e. to find a closed algebra formed by superoperators that contain the Lindbladian $\mathcal{L}_B$ itself.

**Closed algebra**

To find this algebra, we note that $\mathcal{L}_B$ can be expressed in the superoperator-formalism as

$$\mathcal{L}_B(\hat{\rho}') = \left( \sum_{\mathbf{k}, k_z \geq 0} \tilde{\mathcal{L}}_{\mathbf{k}} \right) \circ \hat{\rho}'. \tag{31}$$

Here $\tilde{\mathcal{L}}_{\mathbf{k}}$ is a superoperator act only on modes $\mathbf{k}$ and $-\mathbf{k}$, which is a linear combination of seven quadratic superoperators. We label these superoperators by $\tilde{h}_i^{\mathbf{k}}$, $i = 1, 2, \ldots 7$. They are defined by

$$\tilde{h}_{\mathbf{k}}^1 \circ \hat{\rho}' = \left( \hat{a}_{\mathbf{k}}^\dagger \hat{a}_{\mathbf{k}} + \hat{a}_{-\mathbf{k}} \hat{a}_{-\mathbf{k}}^\dagger \right) \hat{\rho}', \qquad \tilde{h}_{\mathbf{k}}^2 \circ \hat{\rho}' = 2\hat{a}_{\mathbf{k}}^\dagger \hat{a}_{-\mathbf{k}}^\dagger \hat{\rho}',$$

$$\tilde{h}_{\mathbf{k}}^3 \circ \hat{\rho}' = 2\hat{a}_{\mathbf{k}} \hat{a}_{-\mathbf{k}} \hat{\rho}', \qquad \tilde{h}_{\mathbf{k}}^4 \circ \hat{\rho}' = \hat{\rho}' \left( \hat{a}_{\mathbf{k}}^\dagger \hat{a}_{\mathbf{k}} + \hat{a}_{-\mathbf{k}} \hat{a}_{-\mathbf{k}}^\dagger \right),$$

$$\tilde{h}_{\mathbf{k}}^5 \circ \hat{\rho}' = 2\hat{\rho}' \hat{a}_{\mathbf{k}}^\dagger \hat{a}_{-\mathbf{k}}^\dagger, \qquad \tilde{h}_{\mathbf{k}}^6 \circ \hat{\rho}' = 2\hat{\rho}' \hat{a}_{\mathbf{k}} \hat{a}_{\mathbf{k}},$$

$$\tilde{h}_{\mathbf{k}}^7 \circ \hat{\rho}' = \hat{a}_{\mathbf{k}} \hat{\rho}' \hat{a}_{\mathbf{k}}^\dagger + \hat{a}_{-\mathbf{k}} \hat{\rho}' \hat{a}_{-\mathbf{k}}^\dagger.$$

Among these seven superoperators, $\tilde{h}_{\mathbf{k}}^i$, $i = 1, 2, \ldots 6$ represent the (anti-)commutators between the (anti-)Hermitian parts of the effective Hamiltonian and the density matrix $\hat{\rho}$, and the last one represents the recycling term of $\tilde{\mathcal{L}}_{\mathbf{k}}$.

However, only these seven superoperators cannot form a closed algebra, i.e. their commutators are not necessarily linear combinations of themselves. For example, we have

$$[\tilde{h}_{\mathbf{k}}^5, \tilde{h}_{\mathbf{k}}^7] \circ \hat{\rho}' = -2 \left( \hat{a}_{\mathbf{k}} \hat{\rho}' \hat{a}_{-\mathbf{k}}^\dagger + \hat{a}_{-\mathbf{k}} \hat{\rho}' \hat{a}_{\mathbf{k}}^\dagger \right). \tag{32}$$

which is a superoperator that can not be written in the form of $\sum_{i=1}^7 \alpha_i \tilde{h}_{\mathbf{k}}^i$.

Table 1: Commutation relation table of 10 superoperators. The superoperators $\tilde{h}^1, \tilde{h}^2, \tilde{h}^3$ or $\tilde{h}^4, \tilde{h}^5, \tilde{h}^6$ can form two $su(1,1)$ algebra, which is the result of the SU(1,1) dynamical symmetry of Bogoliubov Hamiltonian in closed system.

| | $\tilde{h}^1]$ | $\tilde{h}^2]$ | $\tilde{h}^3]$ | $\tilde{h}^4]$ | $\tilde{h}^5]$ | $\tilde{h}^6]$ | $\tilde{h}^7]$ | $\tilde{h}^8]$ | $\tilde{h}^9]$ | $\tilde{h}^{10}]$ |
|---|---|---|---|---|---|---|---|---|---|---|
| $[\tilde{h}^1,$ | 0 | $2\tilde{h}^2$ | $-2\tilde{h}^3$ | 0 | 0 | 0 | $-\tilde{h}^7$ | $\tilde{h}^8$ | $-\tilde{h}^9$ | $\tilde{h}^{10}$ |
| $[\tilde{h}^2,$ | | 0 | $-4\tilde{h}^1$ | 0 | 0 | 0 | $-2\tilde{h}^8$ | 0 | $-2\tilde{h}^{10}$ | 0 |
| $[\tilde{h}^3,$ | | | 0 | 0 | 0 | 0 | 0 | $2\tilde{h}^7$ | 0 | $2\tilde{h}^9$ |
| $[\tilde{h}^4,$ | | | | 0 | $2\tilde{h}^5$ | $-2\tilde{h}^6$ | $-\tilde{h}^7$ | $-\tilde{h}^8$ | $\tilde{h}^9$ | $\tilde{h}^{10}$ |
| $[\tilde{h}^5,$ | | | | | 0 | $-4\tilde{h}^4$ | $-2\tilde{h}^9$ | $-2\tilde{h}^{10}$ | 0 | 0 |
| $[\tilde{h}^6,$ | | | | | | 0 | 0 | 0 | $2\tilde{h}^7$ | $2\tilde{h}^8$ |
| $[\tilde{h}^7,$ | | | | | | | 0 | $\tilde{h}^6$ | $\tilde{h}^3$ | $\tilde{h}^1+\tilde{h}^4$ |
| $[\tilde{h}^8,$ | | | | | | | | 0 | $\tilde{h}^1-\tilde{h}^4$ | $\tilde{h}^2$ |
| $[\tilde{h}^9,$ | | | | | | | | | 0 | $\tilde{h}^5$ |
| $[\tilde{h}^{10},$ | | | | | | | | | | 0 |

It is thus necessary to introduce three more superoperators $\tilde{h}^i_{\mathbf{k}}$, $i = 8, 9, 10$ to close the algebra. They are

$$\tilde{h}^8_{\mathbf{k}} \circ \hat{\rho}' = \hat{a}^\dagger_{-\mathbf{k}} \hat{\rho}' \hat{a}^\dagger_{\mathbf{k}} + \hat{a}^\dagger_{\mathbf{k}} \hat{\rho}' \hat{a}^\dagger_{-\mathbf{k}},$$
$$\tilde{h}^9_{\mathbf{k}} \circ \hat{\rho}' = \hat{a}_{\mathbf{k}} \hat{\rho}' \hat{a}^\dagger_{-\mathbf{k}} + \hat{a}_{-\mathbf{k}} \hat{\rho}' \hat{a}^\dagger_{\mathbf{k}},$$
$$\tilde{h}^{10}_{\mathbf{k}} \circ \hat{\rho}' = \hat{a}^\dagger_{\mathbf{k}} \hat{\rho}' \hat{a}_{\mathbf{k}} + \hat{a}^\dagger_{-\mathbf{k}} \hat{\rho}' \hat{a}_{\mathbf{k}}.$$

All of the superoperators $\tilde{h}^i_{\mathbf{k}}$, $i = 1, 2, \ldots, 10$ now form a closed algebra,[3] whose commutation relations are listed in Table. 1. Hence these superoperators are generators of the dynamical symmetry group of Lindbladian. In the following of this section, because only particles at momentum $\mathbf{k}$ and $-\mathbf{k}$ are coupled, we will omit the label $\mathbf{k}$ of superoperators. Below we will prove this algebra can map to the algebra of two coupled harmonic oscillators and the corresponding group is isomorphic to Sp$(4, \mathbb{C})$. Before that, we will formally write the exact solution of $\tilde{\mathcal{L}}_B$.

**Exact solution**

Based on this closed algebra structure, the dynamical problem of time-dependent Lindbladian can be solved exactly [53, 54]. First of all, formally solve the master equation $\hat{\rho}$ as $\partial_t \hat{\rho} = \tilde{\mathcal{L}}_B \circ \hat{\rho}$, we can denote the solution as

$$\hat{\rho}(t) = \tilde{\Lambda}(t, t_0) \circ \hat{\rho}(t_0). \tag{33}$$

$\tilde{\Lambda}(t, t_0)$ is a dynamical map from time $t_0$ to time $t$. In general, this mapping is a semi-group which satisfies $\tilde{\Lambda}(t_2, t_0) = \tilde{\Lambda}(t_2, t_1)\tilde{\Lambda}(t_1, t_0)$ but don't satisfy the unitary condition under Lindblad time evolution. Thanks for the closed algebra, the Lindbladian can be written as a linear combination of the dynamical symmetry group generators $\tilde{\mathcal{L}} = \sum_{l=1}^{7} u_l(t)\tilde{h}^l$ hence we can parametrize the dynamical map by these ten generators,

$$\tilde{\Lambda}(t, t_0) = \prod_{i=1}^{10} e^{g_i(t)\tilde{h}^i}. \tag{34}$$

---

[3]In fact, the structure of the master equation can further restrict these 10 elements into seven: $\tilde{\mathcal{H}}^0_{\mathbf{k}} = \tilde{h}^1_{\mathbf{k}} - \tilde{h}^4_{\mathbf{k}}$, $\tilde{\mathcal{H}}^1_{\mathbf{k}} = \tilde{h}^2_{\mathbf{k}} - \tilde{h}^6_{\mathbf{k}}$, $\tilde{\mathcal{H}}^2_{\mathbf{k}} = \tilde{h}^3_{\mathbf{k}} - \tilde{h}^5_{\mathbf{k}}$, $\tilde{\mathcal{L}}^0_{\mathbf{k}} = \tilde{h}^1_{\mathbf{k}} + \tilde{h}^4_{\mathbf{k}} - 2\tilde{h}^7_{\mathbf{k}}$, $\tilde{\mathcal{L}}^1_{\mathbf{k}} = \tilde{h}^2_{\mathbf{k}} + \tilde{h}^6_{\mathbf{k}} - 2\tilde{h}^8_{\mathbf{k}}$, $\tilde{\mathcal{L}}^2_{\mathbf{k}} = \tilde{h}^3_{\mathbf{k}} + \tilde{h}^5_{\mathbf{k}} - 2\tilde{h}^9_{\mathbf{k}}$, $\tilde{\mathcal{L}}^3_{\mathbf{k}} = \tilde{h}^1_{\mathbf{k}} + \tilde{h}^4_{\mathbf{k}} - 2\tilde{h}^{10}_{\mathbf{k}}$.

Substitute this equation back to the master equation, we can get

$$\partial_t \prod_{i=1}^{10} e^{g_i(t)\tilde{h}^i} = \sum_{l=1}^{7} u_l(t)\tilde{h}^l \prod_{j=1}^{10} e^{g_j(t)\tilde{h}^j},\tag{35}$$

where $u_l(t)$ are time-dependent parameters defined by Hamiltonian. Solving the dynamics of the time-evolution operator is equivalent to solving the complex functions $g_i(t)$. Right multiply the inverse of $\tilde{\Lambda}(t, t_0)$ at both sides of eq. (35) and write the explicit expression of time derivative, we can obtain

$$\sum_{m=1}^{10} \partial_t g_m(t) \prod_{i=1}^{m} e^{g_i(t)\tilde{h}^i} \tilde{h}^m e^{-g_i(t)\tilde{h}^i} = \sum_{l=1}^{7} u_l(t)\tilde{h}^l.\tag{36}$$

Using Baker-Campbell-Hausdorf formula [55] and commutation relation in Table. 1, we can formally write L.H.S of eq. (36) as:

$$\sum_{m,n} \partial_t g_m(t) \eta_{mn} \tilde{h}^n = \sum_{l=1}^{7} u_l(t)\tilde{h}^l,\tag{37}$$

where the $\eta_{mn}$ are analytic functions of $g$. Considering the linear independent property of ten generators, we can obtain a set of coupled first-order differential equations. In consequence, by solving these coupled differential equations for $g(t)$, we will get the exact solution of the dynamics governed by time-dependent Lindbladian. In practice, obtaining an analytical solution for a general time dependence $u_l(t)$ is difficult. However, eq. (37) offers a method to numerically obtain the time evolution of the density matrix, which is similar to the evolution of a Gaussian state under a quadratic Hamiltonian [46, 56].

In the following, we show that the superoperators $\tilde{h}^i$ form an algebra of $sp(4, \mathbb{C})$ by mapping them to a coupled 2-mode harmonic oscillator. We further generalize this result to an $n$-mode quadratic Lindblad bosonic system, in which the superoperators form a closed algebra of $sp(2n, \mathbb{C})$.

**Map to harmonic oscillators**

Even though the commutation relations between $\tilde{h}^i_{\mathbf{k}}$ seem complicated as shown in Table. 1, it is worth noting that they only consist of quadratic superoperators. A natural question is then whether there is an isolated system that has the same algebra structure. Indeed, we find that if we consider two coupled harmonic oscillators with ladder operators $a$ and $b$, all the superoperators $\tilde{h}^i$ can be mapped to linear combinations of quadratic forms of $a, a^\dagger, b, b^\dagger$. In Table. 2, we give the explicit correspondence of this mapping, and it is not difficult to show that the mapping preserves commutation relations, i.e. it is indeed an isomorphism. The isomorphism greatly the structure of the algebra. As an example, we will show in the following that the quadratics of ladder operators $\hat{a}$ and $\hat{b}$ form the algebra of symplectic group $\text{Sp}(4, \mathbb{C})$.

**$Sp(2n, \mathbb{C})$ dynamical symmetry group**

To show this, we prove a general result, i.e. the quadratic forms of ladder operators $a_n$ in an $n$-mode harmonic oscillators ($n = 1, 2, \ldots$) have symplectic $\text{Sp}(2n, \mathbb{C})$ dynamical symmetry. A

Table 2: Operators in coupled harmonic oscillator system and superoperators in dissipative Bose gas system after Bogoliubov approximation with momentum $\mathbf{k}$. These 10 operators and superoperators have the same commutation relation in Table. 1 and they can form the algebra $sp(4, \mathbb{C})$.

|          | Harmonic Oscillator | Bogoliubov Lindbladian |
|----------|---------------------|------------------------|
| $h^1$    | $(a^\dagger a + a a^\dagger)/2$ | $(\hat{a}_\mathbf{k}^\dagger \hat{a}_\mathbf{k} + \hat{a}_{-\mathbf{k}} \hat{a}_{-\mathbf{k}}^\dagger)\circ$ |
| $h^2$    | $a^\dagger a^\dagger$ | $(2\hat{a}_\mathbf{k}^\dagger \hat{a}_{-\mathbf{k}}^\dagger)\circ$ |
| $h^3$    | $aa$ | $(2\hat{a}_\mathbf{k} \hat{a}_{-\mathbf{k}})\circ$ |
| $h^4$    | $(b^\dagger b + b b^\dagger)/2$ | $\circ(\hat{a}_\mathbf{k}^\dagger \hat{a}_\mathbf{k} + \hat{a}_{-\mathbf{k}} \hat{a}_{-\mathbf{k}}^\dagger)$ |
| $h^5$    | $b^\dagger b^\dagger$ | $\circ(2\hat{a}_\mathbf{k}^\dagger \hat{a}_{-\mathbf{k}}^\dagger)$ |
| $h^6$    | $bb$ | $\circ(2\hat{a}_\mathbf{k} \hat{a}_{-\mathbf{k}})$ |
| $h^7$    | $ab$ | $\hat{a}_\mathbf{k} \circ \hat{a}_\mathbf{k}^\dagger + \hat{a}_{-\mathbf{k}} \circ \hat{a}_{-\mathbf{k}}^\dagger$ |
| $h^8$    | $a^\dagger b$ | $\hat{a}_{-\mathbf{k}}^\dagger \circ \hat{a}_\mathbf{k}^\dagger + a_\mathbf{k}^\dagger \circ a_{-\mathbf{k}}^\dagger$ |
| $h^9$    | $ab^\dagger$ | $\hat{a}_\mathbf{k} \circ \hat{a}_{-\mathbf{k}}^\dagger + \hat{a}_{-\mathbf{k}} \circ \hat{a}_\mathbf{k}^\dagger$ |
| $h^{10}$ | $a^\dagger b^\dagger$ | $\hat{a}_\mathbf{k}^\dagger \circ \hat{a}_\mathbf{k} + \hat{a}_{-\mathbf{k}}^\dagger \circ \hat{a}_{-\mathbf{k}}$ |

generic quadratic form is given by is given by

$$
\hat{M} = \frac{1}{2} \begin{pmatrix} a_1^\dagger, & \ldots, & a_n^\dagger, & a_1, & \ldots, & a_n \end{pmatrix} \begin{pmatrix} B & A \\ A^\mathrm{T} & C \end{pmatrix} \begin{pmatrix} a_1^\dagger \\ \vdots \\ a_n^\dagger \\ a_1 \\ \vdots \\ a_n \end{pmatrix}.
$$

To make the coefficient matrices unique, we require $B^\mathrm{T} = B$ and $C^\mathrm{T} = C$.

Note that the coefficient matrix may be written as

$$
\begin{pmatrix} B & A \\ A^\mathrm{T} & C \end{pmatrix} = \begin{pmatrix} A & -B \\ C & -A^\mathrm{T} \end{pmatrix} \begin{pmatrix} & I \\ -I & \end{pmatrix} \equiv M\Omega, \tag{38}
$$

with $M$ satisfying

$$
\Omega M + M^\mathrm{T}\Omega = 0, \tag{39}
$$

which is the generator for Lie group $\mathrm{Sp}(2n, \mathbb{C})$.

We now define new operators

$$
b_i = \begin{cases} a_i, & i \in \{1, 2, \ldots, n\}, \\ a_{i-n}, & i \in \{n+1, \ldots, 2n\}, \end{cases} \tag{40}
$$

$$
b^i = \begin{cases} a_i^\dagger, & i \in \{1, 2, \ldots, n\}, \\ a_{i-n}, & i \in \{n+1, \ldots, 2n\}, \end{cases} \tag{41}
$$

which are related by $b_i = \Omega_{ij} b^j$ and $b^i = \Omega^{ij} b_j$ with matrices

$$
\Omega_{ij} = \begin{cases} \delta_{i, j-n}, & i \in \{1, 2, \ldots, n\}, \\ -\delta_{i, j+n}, & i \in \{n+1, \ldots, 2n\}, \end{cases} \tag{42}
$$

$$
\Omega^{ij} = \begin{cases} -\delta_{i, j-n}, & i \in \{1, 2, \ldots, n\}, \\ \delta_{i, j+n}, & i \in \{n+1, \ldots, 2n\}. \end{cases} \tag{43}
$$

We thus have the following useful commutation relations

$$\left[b_i, b^j\right] = \delta_i^j, \qquad \left[b^i, b_j\right] = -\delta_j^i, \tag{44}$$

$$\left[b^i, b^j\right] = \Omega^{ij}, \qquad \left[b_i, b_j\right] = -\Omega_{ij}. \tag{45}$$

With all these definitions, the generic quadratic form in Eq. (38) can then be expressed as

$$\hat{M} = \frac{1}{2} b^i M_i^j b_j, \tag{46}$$

with the upper (lower) index representing the column (row) index. The above formula gives a one-to-one mapping between the quadratic form and the Lie algebra $sp(2n, \mathbb{C})$. More importantly, it can be proved (Appendix A)

$$\left[\hat{M}, \hat{N}\right] = \frac{1}{2} b^i \left[M, N\right]_i^j b_j, \tag{47}$$

which implies that the mapping is an isomorphism.

Now we prove $n$-mode quadratic bosonic Hamiltonian has an $\mathrm{Sp}(2n, \mathbb{C})$ dynamical symmetry, meanwhile, in the last subsection we proved our Bogoliubov Lindbladian is isomorphic to coupled 2-mode harmonic oscillator. In consequence, we can conclude that Bogoliubov Lindbladian has an $\mathrm{Sp}(4, \mathbb{C})$ dynamical symmetry.

Furthermore, our results are not only restricted to Bogoliubov Lindbladian. Because quadratic operators in $2n$-mode coupled harmonic oscillator have the same commutation relation with the superoperators in $n$-mode quadratic Lindbladian, we can conclude that quadratic Lindbladian which is constituted by $n$-mode bosons has $\mathrm{Sp}(2n, \mathbb{C})$ dynamical symmetry. This result will be useful for the further analytical study of open systems.

# 6 Dissipative Gross-Pitaevskii equation and hydrodynamic theory

In closed systems, weakly interacting Bose gas can also be treated using other theoretical approaches such as the Gross-Pitaevskii equation and the hydrodynamic theory [49, 50, 57]. In this section, we generalize these two descriptions to open systems with weak two-body losses. We derive the dissipative versions of the Gross-Pitaevskii equation and hydrodynamic equations based on the Keldysh path integral formalism.

**Keldysh formalism**

We start from the master equation $\partial_t \hat{\rho} = \mathcal{L}\hat{\rho}$ of interacting bosons subject to two-body losses in real space,

$$\mathcal{L}\hat{\rho} = \frac{1}{i}[\hat{H}, \hat{\rho}] - \frac{\gamma}{2} \int_\mathbf{r} \{\hat{\psi}^\dagger(\mathbf{r})\hat{\psi}^\dagger(\mathbf{r})\hat{\psi}(\mathbf{r})\hat{\psi}(\mathbf{r}), \hat{\rho}\} + \mathcal{J}\hat{\rho}, \tag{48}$$

where $\hat{\psi}(\mathbf{r})$ is the bosonic annihilation operator at position $\mathbf{r}$, and

$$\hat{H} = \int_\mathbf{r} \hat{\psi}^\dagger(\mathbf{r})\left(-\frac{\nabla^2}{2m}\right)\hat{\psi}^\dagger(\mathbf{r}) + \frac{g}{2} \int_\mathbf{r} \hat{\psi}^\dagger(\mathbf{r})\hat{\psi}^\dagger(\mathbf{r})\hat{\psi}(\mathbf{r})\hat{\psi}(\mathbf{r}), \tag{49}$$

is Hermitian Hamiltonian of interacting bosons.

The recycling term $\mathcal{J}\hat{\rho}$ is given by

$$\mathcal{J}\hat{\rho} = \gamma \int_\mathbf{r} \hat{\psi}(\mathbf{r})\hat{\psi}(\mathbf{r})\hat{\rho}\,\psi^\dagger(\mathbf{r})\hat{\psi}^\dagger(\mathbf{r}). \tag{50}$$

Using the keldysh path-integral representation, we introduce fields $\psi_\pm$ and $\bar{\psi}_\pm$ living respectively on the time contour $\mathcal{C}_+$ and $\mathcal{C}_-$, where time runs from $-\infty$ to $+\infty$ and then back to $-\infty$ in the closed-contour $\mathcal{C}_+ \cup \mathcal{C}_-$. Then we can write partition function $Z = \text{Tr}(\rho(t))$ of eq. (48),

$$Z = \int \mathcal{D}[\psi_+, \bar{\psi}_+, \psi_-, \bar{\psi}_-] e^{iS[\psi]}, \tag{51}$$

where we omit the initial condition and the Lindblad-Keldysh action [58] is

$$
\begin{aligned}
S = &\int dt d\mathbf{r} \sum_{\eta=\pm} (-1)^{s_\eta} (\bar{\psi}_\eta(\mathbf{r}) i \partial_t \psi_\eta(\mathbf{r}) - H(\bar{\psi}_\eta(\mathbf{r}), \psi_\eta(\mathbf{r}))) \\
&+ \frac{i}{2}\gamma \int dt d\mathbf{r} \sum_{\eta=\pm} \bar{\psi}_\eta(\mathbf{r})\bar{\psi}_\eta(\mathbf{r})\psi_\eta(\mathbf{r})\psi_\eta(\mathbf{r}) \\
&- i\gamma \int dt d\mathbf{r} \ \ \psi_+(\mathbf{r})\psi_+(\mathbf{r})\bar{\psi}_-(\mathbf{r})\bar{\psi}_-(\mathbf{r}),
\end{aligned}
\tag{52}
$$

where time runs from $-\infty$ to $+\infty$, and $s_\eta = 0$ for $\eta = +$, $s_\eta = 1$ for $\eta = -$.

## Saddle-point approximation

We consider the case that almost all particles occupy the ground state energy level, which means $N \approx N_0 \gg 1$. In the closed system, we can assume the system is always in a coherent state, and the condensate wavefunction can be found by the saddle-point equation. Similarly, here we consider the losses process to be weak and slow so that we can still take the coherent state assumption. As a result, we can take the saddle-point equation in an open system,

$$\frac{\delta S}{\delta \bar{\psi}_\pm} = 0, \frac{\delta S}{\delta \psi_\pm} = 0, \tag{53}$$

then we obtain

$$
\begin{aligned}
i\partial_t \psi_+ - \left( -\frac{\nabla^2}{2m}\psi_+ + g_c \bar{\psi}_+ \psi_+ \psi_+ \right) &= 0, \\
-i\partial_t \bar{\psi}_+ - \left( -\frac{\nabla^2}{2m}\bar{\psi}_+ + g_c \bar{\psi}_+ \psi_+ \bar{\psi}_+ \right) - 2i\gamma \psi_+ \bar{\psi}_- \bar{\psi}_- &= 0, \\
-i\partial_t \psi_- + \left( -\frac{\nabla^2}{2m}\psi_- + g_c^* \bar{\psi}_- \psi_- \psi_- \right) - 2i\gamma \psi_+ \psi_+ \bar{\psi}_- &= 0, \\
i\partial_t \bar{\psi}_- + \left( -\frac{\nabla^2}{2m}\bar{\psi}_- + g_c^* \bar{\psi}_- \psi_- \bar{\psi}_- \right) &= 0,
\end{aligned}
\tag{54}
$$

where the complex interacting strength $g_c = g - i\gamma$.

## Non-Hermitian Gross-Pitaevskii equation

The regularization of Keldysh action requires the relation $\psi_+ = \psi_-, \bar{\psi}_+ = \bar{\psi}_-$ [59], combining with the saddle-point equations eq. (54), we will obtain the Gross-Pitaevskii equation under the two-body losses ($\psi_+ = \psi$),

$$i\partial_t \psi - \left( -\frac{\nabla^2}{2m}\psi + g_c|\psi|^2\psi + V\psi \right) = 0. \tag{55}$$

This equation substitutes the coupling parameter $g$ to the complex version $g - i\gamma$ in the conventional Gross-Pitaevskii equation. However, due to the particle loss, the solution of this dissipative Gross-Pitaevskii equation becomes quite different from the conventional solution for a closed system. For example, when $V = 0$, it is easy to show that we have a plane-wave-like solution given by

$$\psi_0(\mathbf{r}) = \sqrt{n_0} e^{-i\frac{g_c}{2\gamma}\log(1+2\gamma n_0 t)} . \tag{56}$$

And in finite momentum we can also obtain an exact solution,

$$\psi_{\mathbf{p}}(\mathbf{r}) = \sqrt{n_0} e^{i(\mathbf{p}\cdot\mathbf{r}-p^2 t/2)} e^{-i\frac{g_c}{2\gamma}\log(1+2\gamma n_0 t)} .$$

Then we can solve the excitation spectrum of this dissipative Gross-Pitaevskii equation by inserting small perturbation $\psi = \psi_0 + \delta\psi$ and its complex conjugate and expanding to the linear order in $\delta\psi$:

$$i\partial_t \delta\psi = -\frac{1}{2}\nabla^2 \delta\psi + 2g_c|\psi_0|^2 \delta\psi + (g - i\gamma)\psi_0^2 \delta\psi^* ,$$

$$i\partial_t \delta\psi^* = \frac{1}{2}\nabla^2 \delta\psi^* - 2g_c^*|\psi_0|^2 \delta\psi^* - (g + i\gamma)\psi_0^{*2}\delta\psi .$$

To solve this equation, we may apply a unitary transformation $\delta\phi = e^{i\frac{g}{2\gamma}\log(1+2\gamma n_0 t)}\delta\psi$ which leads to a simplified equation in the "rotating wave frame",

$$i\partial_t \delta\phi = -\frac{1}{2}\nabla^2 \delta\phi + (2g_c - g)n\delta\phi + g_c n\delta\phi^* ,$$

$$i\partial_t \delta\phi^* = \frac{1}{2}\nabla^2 \delta\phi^* - (2g_c^* - g)n\delta\psi^* - g_c^* n\delta\phi ,$$

with $n = n_0(1 + 2\gamma n_0 t)^{-1}$. Now if we consider perturbations with momentum $\mathbf{k}$ and naively treat $n(t)$ as time-invariant, we may find the eigenfrequencies of the above equations are

$$\omega_{\mathbf{k}} = -2i\gamma n \pm \sqrt{\epsilon_{\mathbf{k}}^2 + 2gn\epsilon_{\mathbf{k}} - \gamma^2 n^2} , \tag{57}$$

which coincide with the values we previously obtained using the Bogoliubov approximation. As a result, we can also get the phase diagram by solving this dissipative Gross-Pitaevskii equation.

**Hydrodynamic equation**

In the closed system, fruitful physical consequences can be obtained by solving the hydrodynamic theory of interacting BEC, such as superfluidity, anisotropic expansion, and low-energy modes in a harmonic trap [49, 50, 57]. It is then interesting to construct the hydrodynamic equation with two-body loss. Starting from the time-dependent dissipative Gross-Pitaevskii equation eq. (55), we can derive the hydrodynamic equations which govern the dynamics of a dissipative fluid. By decomposing $\psi = \sqrt{\rho}e^{i\theta}$, the real part and imaginary part give the hydrodynamic equations:

$$\partial_t \rho + \nabla(\rho\mathbf{v}) + 2\gamma\rho^2 = 0 ,$$

$$m\partial_t \mathbf{v} = -\nabla\left(\frac{1}{2m}\frac{1}{\sqrt{\rho}}\nabla^2\sqrt{\rho} + \frac{1}{2}mv^2 + V(\mathbf{r}) + g\rho\right) . \tag{58}$$

The first equation is called the continuity equation, which reflects the conservation of particle numbers. Now two-body losses in interacting BEC bring new term $\gamma\rho^2$ in the continuity equation, which breaks the particle number conservation. And the second equation, the Newton

equation, is same as the conventional one. For a uniform system with $V(\mathbf{r}) = 0$, the uniform solution is

$$\rho(\mathbf{r}, t) = \frac{\rho_0}{1 + 2\gamma t \rho_0},\tag{59}$$

which shows that condensate particles always decay with time and the vacuum is the only true steady state of this system. With the new dissipation term $\gamma \rho^2$, finding the solutions for the general case is challenging, we will leave this for future research.

## 7 Conclusion

In summary, we systematically study the many-body dynamics of weakly interacting Bose gases with two-body losses. It is shown that both the two-body interactions and losses in a cold atomic gas may be described by a complex scattering length $a_c$, which may be controlled via tuning an external laser field. We generalize Bogoliubov approximation to open systems and verify the validity of this approximation by numerical simulating a toy model that has a similar structure. Based on this time-dependent Bogoliubov Lindbladian, we study the quench problem and prove this system has a $\text{Sp}(4, \mathbb{C})$ dynamical symmetry, which is crucial for the exact calculation of quench dynamics. Furthermore, we show a general $n$-mode quadratic Lindbladian of the bosonic system has a dynamical symmetry of $\text{Sp}(2n, \mathbb{C})$, which is useful for the analytical understanding of the dissipative open system. On the other hand, we also generalize the Gross-Pitaevskii equation and hydrodynamic theory to dissipative Bose gases. Hydrodynamic equations have rich interesting solutions in closed systems, we only discuss the solution without external potential in this paper, it would be significant to generalize these solutions to open systems and study the stable property of these results. Finally, from a closed interacting bosonic model to open system, other types dissipation are admitted such as single body loss and pump, particle number form dissipation. It is also interesting to discuss the different properties of these open systems. We hope our work can be helpful for further understanding the non-equilibrium dissipative dynamics in cold atom systems.

## Acknowledgments

We gratefully acknowledge fruitful discussions with Hui Zhai and Xin Chen.

**Funding information** This work is supported by NSFC under Grant No. 12204352.

## A Proof of Eq. (47)

We prove Eq. (47) by direct calculation,

$$
\begin{aligned}
[\hat{M}, \hat{N}] &= \frac{1}{4} \left[ b^i M_i^j b_j, b^l N_l^k b_k \right] = \frac{1}{4} M_i^j N_l^k \left[ b^i b_j, b^l b_k \right] \\
&= \frac{1}{4} M_i^j N_l^k \left( b^i \left[ b_j, b^l \right] b_k + b^i b^l \left[ b_j, b_k \right] + \left[ b^i, b^l \right] b_k b_j + b^l \left[ b^i, b_k \right] b_j \right) \\
&= \frac{1}{4} M_i^j N_l^k \left( \delta_j^l b^i b_k - \Omega_{jk} b^i b^l + \Omega^{il} b_k b_j - \delta_k^i b^l b_j \right)
\end{aligned}
$$

$$= \frac{1}{4}\left(M_i^j N_j^k b^i b_k - M_i^j N_l^i b^l b_j + M_i^j N_l^k \left(\Omega^{il} b_k b_j - \Omega_{jk} b^i b^l\right)\right)$$

$$= \frac{1}{4}\left(b^i [M,N]_i^j b_j + M_i^j N_l^k \left(\Omega^{il} b_k b_j - \Omega_{jk} b^i b^l\right)\right). \tag{A.1}$$

Now we are left with the second term on the R.H.S. To keep going, note Eq. (39) is written as

$$\Omega^{ij} M_j^l + M_j^i \Omega^{jl} = 0, \qquad \Omega_{ij} M_l^j + M_i^j \Omega_{jl} = 0. \tag{A.2}$$

We thus have

$$\begin{aligned}
M_i^j N_l^k \left(\Omega^{il} b_k b_j - \Omega_{jk} b^i b^l\right) &= \Omega^{il} M_i^j N_l^k \Omega_{kq} b^q b_j - \Omega_{jk} M_i^j N_l^k \Omega^{ls} b^i b_s \\
&= -\Omega^{il} M_i^j \Omega_{lk} N_q^k b^q b_j + \Omega_{jk} M_i^j \Omega^{kl} N_l^s b^i b_s \\
&= -M_i^j N_q^i b^q b_j + M_i^j N_j^s b^i b_s \\
&= b^i [M,N]_i^j b_j. \tag{A.3}
\end{aligned}$$

Combing Eq. (A.1) and Eq. (A.3), we then proved Eq. (47) in the main text.

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
