# Peer review of "Weakly interacting Bose gas with two-body losses"

_SciPost Physics, doi:SciPost Phys. 16, 116 (2024)_

## Round 1 · Referee Report · Anonymous · 2023-6-16

Report
The authors have conducted a study on the dynamics of weakly interacting Bose gases with two-body losses. This research is highly relevant to cold atom experiments and addresses new theoretical frontiers, such as open systems and non-Hermitian systems. Specifically, the authors demonstrate that optical Feshbach resonance can effectively control inelastic scattering, resulting in a complex scattering length. They employed numerical solutions of the Lindblad equation and the Bogliubov approximation to analyze the system dynamics. Additionally, the authors investigated the dynamical symmetry of this system. Taking into account all of the above, I believe this paper is timely and provides a comprehensive understanding of the subject matter. The writing is clear and comprehensible. Therefore, I would like to recommend this publication for Scipost. Furthermore, I have a few suggestions for the authors:
(1) Providing a physical explanation of c_theta in Eq (20) would greatly assist readers in comprehending the physical implications of a complex scattering length, particularly when comparing it to a negative scattering length in closed system. This clarification would enhance the understanding of the phenomenon described.
(2) A clarification is required regarding the lack of change in gamma_b while the bare interaction g_b becomes g, as observed from Eq. (16) to (17). There must be some underlying considerations for this discrepancy. Hence, an explanation is necessary to address this inconsistency.
(3) By employing the dynamical symmetry, the system dynamics can be described by a set of first-order differential equations, as illustrated in Eq. (37). It would be valuable to investigate whether it is possible to analytically solve this set of equations. Exploring potential analytical solutions would further enhance the understanding of the system's behavior and contribute to the depth of the analysis.

---

## Round 2 · Referee Report · Anonymous (Referee 1) · 2024-2-7

Strengths

1- Given the recent advances in non-Hermitian and open quantum systems, the results presented in the paper are timely. 2- The results in the paper are robust and comprehensive. 3- Their findings provide valuable insights. Subsequent works focusing on three-body and four-body systems would be straightforward.

Report

The authors have addressed my questions and concerns in a scientific manner. I believe the current manuscript meets the criteria for publication in SciPost. Therefore, I recommend its publication.

---

## Round 2 · Referee Report · Anonymous (Referee 2) · 2024-3-5

Report

The authors study the many-body dynamics of a weakly interacting Bose gas characterized by dissipation in the form of two-body losses. They discuss how inelastic collisions caused by two-body losses lead to a complex scattering length, which can be experimentally tuned via an optical Feshbach resonance. They generalize the Bogoliubov approximation to such open systems. They confirm the validity of this approximation by comparing its results to those of a full numerical calculation based on a toy model which has the same main features (interactions and dissipation). Furthermore, they explore the quench dynamics and dynamical symmetry of these systems. This work is theoretically novel and topically relates to current cold-atom experiments on open systems. The writing is clear and the derivations are comprehensive. I believe the newest version ("v2") of the manuscript meets the publishing criteria of SciPost Physics. Thus, I recommend it for publication.

---

## Round 2 · Referee Report · Anonymous (Referee 3) · 2024-3-8

Report

In the present manuscript the authors explore many-body dynamics in open quantum systems by introducing atomic losses to a ultracold bosonic gas near a optical Feshbach resonance. Essentially, this approach derived by introducing a non-Hermitian absorbing potential that ultimately led to an complex scattering length. Using this model the authors explore a number of characterizations of the open quantum system and performed analysis that led to a number of very interesting results, in particular the derivation of the dissipative GP equation. This manuscript easly satisfy the criterial of journals like Phys. Rev. A and I would like to recommend this manuscript for publication in SciPost. I would like, however, to suggest the authors consider the following points:

1) Please, revise the manuscript for grammar issues
2) In the first sentence of section 2 the words “two-body losses” and “inelastic collisions” should be swapped
3) Please, cite Refs [39-41] after Eq. (1)
4) In figure 1 (and perhaps throughout the manuscript) Im(1/a_c) is shown as a positive quantity. The imaginary part of the complex scattering length is always negative.
5) Below Eq. (8), I’m unsure what the authors meant by “decaying to the environment”. Please, be more precise.
6) Near Eq. (20), please, define explicitly the relation a_c=a_r-i*a_i for the complex scattering length

---

## Round 2 · Author Response

We sincerely appreciate the reviewer's careful reading and positive evaluation of our manuscript.
In the following, we address his/her comments and suggestions.
(1) “Providing a physical explanation of c_theta in Eq (20) would greatly assist readers in comprehending the physical implications of a complex scattering length, particularly when comparing it to a negative scattering length in closed system. This clarification would enhance the understanding of the phenomenon described.”
In dilute Bose gas without dissipation, the LHY correction of the ground state energy becomes ill-defined for a negative scattering length. This corresponds to an instability in the dynamic evolution, where the quantum depletion grows to a non-perturbative value. In comparison, the loss rate for dissipative Bose gas (eq. (20) in the manuscript) also becomes ill-defined for arg(1/a_c)>\pi/6. This also reflects the same instability of exponentially growing Bogoliubov modes because of the attractive interaction. In addition, the fact that the transition argument of 1/a_c is not \pi/2 illustrates that the two-particle loss also helps suppress the increase of these Bogoliubov modes.
We thank the referee for the kind suggestion and added an explanation paragraph in the revised manuscript (page 9, the end of chapter 3).
(2)“A clarification is required regarding the lack of change in gamma_b while the bare interaction g_b becomes g, as observed from Eq. (16) to (17). There must be some underlying considerations for this discrepancy. Hence, an explanation is necessary to address this inconsistency.”
The inconsistency arises from the subtlety of the renormalization of g_b and gamma_b. Recall that in the conventional derivation of the LHY correction of non-dissipative Bose gas (see e.g. C. J. Pethick, H. Smith, Bose–Einstein condensation in dilute gases), the momentum summation of the zero-point energies of Bogoliubov modes also diverges while incorporating a renormalized interacting strength. The divergence arises from the fact that the renormalized interaction is only valid for small momenta. To resolve this issue, one can introduce an intermediate momentum cutoff \Lambda and then consider an effective interaction with second-order processes in the mean-field energy component. Consequently, we can expand the bare interaction strength g_b to the second order of g \Lambda, resulting in a finite LHY correction.
A similar method is implemented in the calculation of the loss rate formula. Instead of an expansion of g_b, we further introduce an expansion of gamma_b (eq.(18) & (19) in the revised version). This cures the divergent momenta summation of eq. (17), resulting in a finite loss rate.
This subtlety in the renormalization calculation is not stated clearly in the first version of our manuscript. We thank the referee for pointing out this which helps us improve the manuscript’s readability.
(3) “By employing the dynamical symmetry, the system dynamics can be described by a set of first-order differential equations, as illustrated in Eq. (37). It would be valuable to investigate whether it is possible to analytically solve this set of equations. Exploring potential analytical solutions would further enhance the understanding of the system's behavior and contribute to the depth of the analysis.”
Indeed, it is possible to find analytical solutions of the first-order differential equations (eq. (37)), which would greatly help to understand the many-body dynamical behavior of the dissipative Bose gases. Yet since it is a challenging task to find the analytical solution for 7 coupled ODEs, we regretfully decide that this direction is beyond the scope of the current work. In the revised manuscript, we have briefly mentioned this interesting possibility and hope it will stimulate future studies on the exact many-body dynamics of dissipative Bose gases.

---

## Round 2 · List of Changes

1. In page 7 and page 8, we have replaced the bare interaction parameters \gamma_b and g_b in Eq. (15-17) with the renormalized interaction \gamma and g.

  2. In page 8, we have added a paragraph to explain the reason for replacing the decay rate \gamma in the mean field term with the effective decay rate \tilde{\gamma} which is corresponding to the second order process from Eq17 to Eq 20.

  3. In page 9, we have added a paragraph to explain the physical effect of the particle number decay rate in Eq 20 at the end of section 3.

  4. In page 14, we have added a paragraph to illustrate that this closed algebra can provide great convenience in numerical operations and a new reference [59] “ C. Weedbrook, S. Pirandola, R. Garc ́ıa-Patr ́on, N. J. Cerf, T. C. Ralph, J. H. Shapiro and S. Lloyd, Gaussian quantum information, Reviews of Modern Physics 84(2), 621 (2012)” has been cited.

  5. We have also updated our funding information.

---

## Round 3 · Referee Report · Anonymous (Referee 3) · 2024-3-27

Report

I appreciate the author's effort to address my comments/questions. I'm satisfied with all the answers and I recommend to manuscript to be published at SciPost as it is.

---

## Round 3 · Author Response

We sincerely appreciate all the reviewers’ careful reading and positive evaluation of our manuscript.
In the following, we address reviewer 3’s comments and suggestions.
(1)"Please, revise the manuscript for grammar issues"
We apologize for the grammatical issues in the article and appreciate the reviewer's suggestions.
We have carefully checked the grammar issues in the article and made corrections in the new version.

(2)"In the first sentence of section 2 the words “two-body losses” and “inelastic collisions” should be swapped"

We have made the correction in the revision as requested by the reviewer. We appreciate the reviewer's careful reading and rigorous logic.

(3) "Please, cite Refs [39-41] after Eq. (1)"

The reference has been added.

(4) "In figure 1 (and perhaps throughout the manuscript) Im(1/a_c) is shown as a positive quantity. The imaginary part of the complex scattering length is always negative."

As pointed by the referee, the imaginary part of the scattering length must be negative. Thus by our definition a_c = a_r + i*a_i with a_i < 0 , the imaginary part of 1/a_c = (a_r - i*a_i)/(a_r^2 + a_i^2) is positive.

(5) "Below Eq. (8), I’m unsure what the authors meant by “decaying to the environment”. Please, be more precise."

By decaying into the environment, we mean the atoms that go through inelastic collisions are no longer confined by the trapping potential.
We appreciate that the referee points out this unclear expression and we have added an explanation in the revised manuscript.

(6) "Near Eq. (20), please, define explicitly the relation a_c=a_r-i*a_i for the complex scattering length"

We define the relation as a_c=a_r + i*a_i with a_i a negative real number. We appreciate that the reviewer point out this confusion and we have added an explicit definition near Eq.(20).

---

## Round 3 · List of Changes

1. swap "inelastic collisions between particles" and "two-body losses" in section 2.

  2. cite Refs [39-41] after Eq. (1)

  3. Below Eq.(8), change "This master equation describes Bose atoms $\hat{a}$ interacting with each other at a bare coupling constant $g_b$ while decaying to the environment at a bare coupling constant $\gamma_b$ though a two-particle losses process." to "This master equation describes the dynamics of Bose atoms, denoted by $\hat{a}$, as they interact with each other through a bare coupling constant $g_b$ while decaying to the environment (i.e. no longer confined by the trapping potential) via a two-body losses process characterized by a bare coupling constant $\gamma_b$."

  4. Below Eq.(20), change "Here $\theta$ is defined as $\theta=-\arg(a_c)=-\arg(a_r+ia_i)\in[0,\pi]$." to "Here, $\theta$ is defined as $\theta=-\arg(a_c) =-\arg(a_r + ia_i) \in[0,\pi]$, where $a_c$ is expressed as two real parameters: $a_c=a_r+ia_i$."

5.A series of grammar issues have been corrected.

---

## Editorial Decision

published